# Functionalization with a Polyphenol-Rich Pomace Extract Empowers a Ceramic Bone Filler with In Vitro Antioxidant, Anti-Inflammatory, and Pro-Osteogenic Properties

**DOI:** 10.3390/jfb12020031

**Published:** 2021-05-05

**Authors:** Giorgio Iviglia, Elisa Torre, Clara Cassinelli, Marco Morra

**Affiliations:** Nobil Bio Ricerche srl, Via Valcastellana 26, 14037 Portacomaro, AT, Italy; etorre@nobilbio.it (E.T.); ccassinelli@nobilbio.it (C.C.); mmorra@nobilbio.it (M.M.)

**Keywords:** bone filler, peri-implantitis, polyphenols, liquid chromatography, antioxidant, anti-inflammatory

## Abstract

Oral diseases and periodontitis in particular are a major health burden worldwide, because of their association with various systemic diseases and with conditions such as peri-implantitis. Attempts have been made over the years to reverse bone loss due to the host disproportionate inflammatory response and to prevent failure of dental implants. To this end, the use of biomaterials functionalized with molecules characterized by anti-inflammatory and antioxidant properties could represent a new frontier for regenerating functional periodontal tissues. In this study, a new ceramic granulated biomaterial, named Synergoss Red (SR), functionalized with a polyphenolic mixture extracted from pomace of the Croatina grape variety, is introduced. Following a preliminary in-depth characterization of the extract by HPLC analysis and of the biomaterial surface and composition, we performed evaluations of cytocompatibility and a biological response through in vitro assays. The anti-inflammatory and antioxidant properties of the identified phenolic molecules contained in SR were shown to downregulate inflammation in macrophages, to stimulate in osteoblast-like cells the expression of genes involved in deposition of the early bone matrix, and to mitigate bone remodeling by decreasing the RANKL/OPG ratio. Thanks to its cytocompatibility and assorted beneficial effects on bone regeneration, SR could be considered an innovative regenerative approach in periodontal therapy.

## 1. Introduction

In the last decades, the development of grafting materials has aroused great interest [1]. In particular, synthetic ceramic materials such as tricalcium phosphate (βTPC) and hydroxyapatite (HA) have been widely used thanks to their good reproducibility, biocompatibility, and non-immunogenicity, in addition to their high similarity to components of the native bone mineral phase [2,3]. The mode of action of biomaterials currently available on the market for bone grafting applications mainly relies on a mechanical support and is thus limited to providing a functional scaffold for cell adhesion. Coupling bone grafting and titanium implants is still one of the best available solutions to replace bone and dental tissue loss [4]. Tooth replacement with titanium dental implants is currently a routine procedure with a success rate around 98% [4]. However, diseases resulting from failures of dental implants are still reported [5]; in fact, recent studies found peri-implantitis to occur in 2.7% to 47.1% of implants [6].

The pathogenesis of periodontal disease is mediated by the host inflammatory response to specific peri-implantitis-associated bacteria [7,8], despite the fact that they are also found in healthy patients, thus suggesting that specific immune and inflammatory mechanisms are required to trigger pathological consequences [9]. Current studies suggest that peri-implantitis is initiated by an aggressive inflammatory host response to bacterial infections, with induction of reactive oxygen species (ROS) formation [5,10].

The commercially available materials used in periodontal regeneration show a lack of fitness for the prevention or control of oxidative stress caused by a potentially uncontrolled inflammation [1,11].

The common strategies for patients affected by periodontitis include the use of a local debridement aimed at eliminating the residual infected tissue, surface decontamination, and the use of classical bone grafting materials, sometimes in combination with a systemic antibiotic therapy [1,12,13,14]. Unfortunately, these approaches are not particularly effective, because they do not prevent or control the tissue damage derived from oxidative stress [12,13].

In this research work, we describe and propose a solution aimed at filling this gap in the bone filler panorama by coupling a mechanical support with modulation of inflammation. For this purpose, understanding how the immune system and the inflammatory response to surgical, biologic, and implant-related factors are regulated and implicated in the success or failure of dental and, more generally, bone implants is essential to develop new treatment solutions [14].

ROS are oxygen free radicals involved in normal cellular metabolism [10] that, under physiological conditions, are neutralized by antioxidant molecules of the saliva [15]. Under inflammatory conditions, cells of the immune system such as neutrophils and macrophages react by producing ROS, which act both signaling molecules and mediators of inflammation [7]. In patients affected by peri-implantitis, an imbalance of ROS occurs, and is also related to a decreased antioxidant power of saliva, as determined by two antioxidant metabolites, namely reduced uric and ascorbic acid [15]. Oxidative stress is an uncontrolled process characterized by overproduction of ROS, which promote the production of inflammatory cytokines that can act at different levels, from DNA and protein damage to oxidation of enzymes and lipid peroxidation, to cell death [10,16]. An increasing volume of studies confirms the association between oxidative stress and peri-implantitis [5,17,18,19]. The use of antioxidants has been shown to significantly reduce the effect of gingivitis, periodontitis and peri-implantitis, drawing a possible route for the development of new combined biomaterials [20].

Recent studies demonstrated that systemically administered antioxidants have effects on gingivitis, while local delivery of gel formulations in the periodontal pocket has been found effective in the treatment of chronic periodontitis in smoking patients [21,22]. Polyphenols such as quercetin and curcumin, thanks to their anti-inflammatory, antioxidant, immunostimulatory, and antiseptic properties, have been widely used in different biomaterials with the aim of mitigating the material-induced oxidative stress and exerting a sustained anti-inflammatory effect [23,24,25]. Furthermore, polyphenols may contribute to the creation of an unfavorable environment for bacteria; in fact, they have been shown to possess antibacterial properties due to their specific chemical structure [26], which determines selective damage to the bacterial membrane [27], in addition to inhibiting bacterial growth and adhesion to surfaces [28].

Thanks to all of these beneficial properties, polyphenols (Pph) can be used to control inflammation and enhance bone regeneration, thus representing a valuable and useful solution. Biosynthesized by plants, Pphs are a class of chemical compounds that exert several actions involved in the molecular defense mechanisms of the plant kingdom. Among the wide panorama of natural sources of these molecules, extraction from waste materials has aroused great interest for ecological, ethical, and economic reasons [29]. One of the most abundant residuals with a high percentage of polyphenolic content is grape pomace (skin and seeds).

Grape by-products represent approximately 20% of the harvested grapes, and in the last decade, many research groups have been focused on the potential of grape pomace as a valuable source of polyphenols [30,31,32]. Depending on the number of phenol rings that Pph contain and on their radicals, they can be divided into several different classes: phenolic acids, stilbenes, anthocyanins, flavonols, and flavan-3-ols. Among these, anthocyanins and flavanols are the most abundant compounds [33]. The involvement of different classes of polyphenols in pathways that can crosstalk with other transduction signals has been extensively studied, with a wide range of applications from bone and cancer to neurodegenerative disorders [34,35]. The mix of polyphenolic compounds that characterize a specific variety of grape pomace extract allows for the local administration of different polyphenols (procyanidins, quercetin, catechin, epigallocatechin, phenolic acids) showing different antioxidant, anti-inflammatory, and pro-osteogenic properties [33], with induction of mesenchymal stem cell osteogenic differentiation [36].

In this work, a synthetic biphasic calcium phosphate granulated biomaterial, functionalized with a releasable antioxidant extract composed of a mixture of batch-to-batch replicable polyphenolic molecules of the local grape variety *Croatina*, is presented, described, and characterized in its ability to counteract the onset of peri-implantitis and to promote bone regeneration.

## 2. Materials and Methods

### 2.1. Materials

All chemicals were analytical-reagent grade. Ultra-pure (MilliQ) water was used for the preparation of aqueous solutions. All chemicals, namely acetone, acetic acid, Folin-Ciocalteu reagent, 2,2-Diphenyl-1-picrylhydrazyl, sodium carbonate, sodium bisulfite, gallic acid, quercetin 3-glucuronide, quercetin, tannic acid, catechin, malvidin 3-glucoside, epicatechin, procyanidin B2, myricetin, quercitrin, kaempferol, isorhamnetin, rutin, epigallocatechin gallate (EPGCG), caffeic acid, and trans p-cumaroyl tartaric acid were purchased from Sigma-Aldrich (St. Louis, MO, USA). Red grape pomace was purchased from a local wine producer (Croatina grape from ALEMAT, Penango, AT, Italy).

### 2.2. Synthesis of Bone Filler

HA was used for enhancing the mechanical strength of scaffolds, whereas β-TCP was used for its degradability; they were mixed in a percentage of 50 wt.%, respectively, to reach an optimum compromise between the two properties. The ceramic scaffolds were prepared by mixing HA and β-TCP powders (47 wt.%) with the binding agent poly (vinyl alcohol) (3 wt.%), and ultrapure water (50 wt.%) to obtain a ceramic slurry. Dolapix CE 64 was added as a dispersing agent (1 wt.% of the solid load). The polyurethane (PU) sponge impregnation method was used to obtain a macroporous ceramic scaffold [37,38]. A commercial PU sponge slab (45 ppi) of 200 × 200 × 10 mm^3^ was soaked into the ceramic slurry for 90 s, followed by compression along the transverse plane (40 kPa) and left at room temperature for 5 min before repeating the cycle. Impregnation/compression cycles were repeated 3 times. The ceramic-coated PU sponge was left to dry overnight at 37 °C and sintered in a furnace at 1100 °C for 12 h in air (heating rate 5 °C/min), in order to obtain a porous HA/βTCP slab of 200 × 200 × 10 mm^3^ (a volumetric retention of 24% was calculated). The obtained porous slab was ground in a jaw crusher (BB 50, Retsch GmbH, Haan, Germany) and sieved in order to obtain a 300–2000 μm porous grained granulate (bone filler control) (BF).

### 2.3. Preparation of Polyphenol-Rich Pomace Extracts (PRPE)

Grape pomace was received in dry form from the producer and stored at −20 °C under vacuum until the beginning of the extraction process. In order to make it suitable for the extraction process, the grape pomace was first washed with acidified water, dried in a circulating-air oven (37 °C ± 5 °C), and ground in a bladed mill (GM 200, Retsch). The milled grape pomace (300 g) was extracted in 2000 mL of 50:50 acetone:water (*v*/*v*) using an automatic extractor (Micro C TIMATIC, Spello (Pg), Italy). The extraction cycle is fully automatic and alternates a dynamic phase, performed at a programmed pressure, and a static phase in which a forced percolation is generated, which, thanks to the programmable recirculation, ensures a continuous flow of solvent to the interior of the plant matrix, thus avoiding over-saturation. Next, the extracted solution was concentrated under reduced pressure in a rotavap in such a way to eliminate the acetone and to obtain an aqueous extract. The concentrated extract was separated by centrifugation (7000 rpm, 5 min) and the supernatant fraction was used to treat the ceramic granulates.

Batch-to-batch variation was analyzed, and the results allowed us to speculate that no significant variations existed among extracts from the same grape pomace variety (Croatina). In Table 1, the antioxidant power and the phenolic content of 3 different batches of extract from Croatina grape pomace, obtained from 3 different measures for each batch and all from the same producer and same year of production, are reported.

An aging study was conducted on the extract. The results showed that, if maintained in a refrigerator between 2–4 °C, it preserves its antioxidant power and phenolic composition up to 2 years.

### 2.4. Synthesis of the Antioxidant Bone Filler

The antioxidant bone filler, namely, Synergoss Red (SR), was produced by soaking 40 wt.% porous ceramic granulate with 60 wt.% polyphenol-rich pomace extract, followed by a 24 h static evaporation process at 45 °C, which allowed polyphenol molecules (Pph) to be physically adsorbed into the ceramic granulate. Considering the total phenolic content of the PRPE used (TC CR 2002—3.17 mg/mL, Table 1), it is possible to speculate that the amount of polyphenols incorporated in 1 g of ceramic granules was around 4.75 mg.

### 2.5. PRPE Characterization

#### 2.5.1. UHPLC Analysis

PRPE from Croatina was characterized by high performance liquid chromatography (HPLC), with the Shimadzu SLC 40 equipped with the SPD-M40 diode array (3 technical replicates were analyzed) (Shimadzu, Kyoto, Japan). PRPE was filtered through 0.2 µm cellulose acetate filters and analyzed using the Kinetex Biphenyl (100 × 3.0 mm^2^, 2.6 μm) column from Phenomenex (Torrance, CA, USA), operated at 40 °C. The mobile phases consisted of 2% (*v*/*v*) acetic acid in water (MPA), 0.5% acetic acid in water, and acetonitrile (50:50 *v*/*v*) (MPB), and were used in the gradient method reported in Table 2, at a flow rate of 0.4 mL/min and total run time of 17 min.

The injection volume was 3 µL and the diode array operated at the wavelength range 200 to 600 nm. Main Pphs were identified through comparison with reference compounds. The quantitation of each Pph was performed by using calibration curves of the corresponding reference compounds. Gallic acid, catechin, tannic acid, procyanidin B2, epicatechin, epigallocatechin gallate (280 nm), quercetin (370 nm), myricetin (375 nm), quercitrin (349 nm), rutin (355 nm), kaempferol, isorhamnetin (366 nm), caffeic acid (310 nm), trans p-cumaroyl tartaric acid (313 nm), quercetin 3-glucuronide (354 nm), and malvidin-3-glucoside (520 nm) were dissolved in ethanol:water solution at the concentrations of 5, 25, 50, 100, 150, and 200 µg/mL and analyzed with the same method reported above. The quantitation was performed by applying the standard calibration curve for each standard.

#### 2.5.2. Total Phenolic Content of PRPE

The initial phenolic content of Croatina PRPE was evaluated by using the Folin-Ciocalteu (FC) method as previously reported [31,39,40]. The extract was transferred in a 25 mL volumetric flask and diluted 1:50 with distilled water (3 technical replicates). Then, 0.5 g of Folin-Ciocalteu reagent was added and mixed for 5 min, and 1.5 g of 20% anhydrous sodium carbonate (*w*/*v*) solution was added. After 2 h, absorbance was measured at 765 nm by using water as the compensation liquid and a quartz cell (10 mm path length) in a UV-Vis spectrophotometer (PG Instruments Limited). The absorbance value was used to calculate the concentration of polyphenols by using a calibration curve obtained with gallic acid. The results are expressed as mg/mL of gallic acid equivalents (GAE).

#### 2.5.3. Calibration Curve

Ten milligrams of gallic acid were diluted in 10 mL of water to obtain 1 mg/mL of stock solution. Aliquots of stock solution were transferred in a 25 mL volumetric flask and diluted in water to the final concentrations of 0.05 mg/mL, 0.025 mg/mL, 0.01 mg/mL, and 0.005 mg/mL. Each standard solution was prepared according to the procedure described above for the PRPE; the absorbance was measured under the same conditions as for PRPE.

#### 2.5.4. Antioxidant Power of PRPE

The antioxidant power of PRPE (1 mg/mL of GAE) and of Pph released from Synergoss Red was evaluated through the widely used DPPH (2,2-Diphenyl-1-picrylhydrazyl) method [41]. This test measures, through a colorimetric approach, the ability of the test solution to scavenge the DPPH radicals. Briefly, an aliquot of 40 µL of PRPE was added to 1600 µL of water:ethanol 50:50 (*v*/*v*) solution (3 technical replicates). Separately, a DPPH solution (0.1 mg/mL *w*/*v*) in ethanol was prepared and 2 mL of this solution was added to the reaction mixture. Then, the solution was shaken and incubated for 30 min at room temperature in the dark; the absorbance was recorded at 525 nm. The blank solution consisted of a solution of water:ethanol instead of PRPE. The inhibition percentage of the DPPH radicals by the samples was calculated using the following equation:(1)$$\%\;\mathrm{Reduction}=\left(\frac{A_{0}-{\;A}_{1}}{A_{0}}\right)\times 100$$
where A_0_ is the absorbance of the control sample and A_1_ is the absorbance of the test sample.

### 2.6. Surface Analysis

#### 2.6.1. Morphological Characterization (SEM)

Scanning electron microscopy (SEM) analysis was performed to observe the surface morphology of bone fillers. Samples were mounted on aluminum stubs and sputtered with gold at 15 mA for 2 min using an agar sputter coater. The morphology of granules was captured using an EVO MA10 system (Zeiss, Oberkochen, Germany).

#### 2.6.2. ATR-IR

ATR-IR spectra were obtained using a Nicolet iS10 ATRIR spectrometer, produced by Thermo Fisher Scientific (Waltham, MA, USA) and equipped with a diamond crystal. Samples were gently placed on the crystal and kept in place using the specific crimping tool. The experimental setup involved acquisition of 32 scans in the range 500–4000 cm^−1^, both of sample and background, at a resolution of 4 cm^−1^.

#### 2.6.3. XPS

XPS analysis was performed using a Perkin Elmer PHI 5600 ESCA system. The instrument was equipped with a monochromatized Al anode operating at 10 kV and 200 W. The diameter of the analyzed spot was approximately 500 micrometers, and the base pressure, 10^−8^ Pa. The angle between the electron analyzer and the sample surface was 45°. Measurements were performed by pressing granules, in order to make a complete layer on a double-sided adhesive tape, 1 side of which was fixed to the instrument sample holder. The analysis was performed by acquiring wide range survey spectra (0–1000 eV binding energy) and detailed high resolution peaks of relevant elements. The quantification of elements was accomplished using the software and sensitivity factors supplied by the manufacturer.

#### 2.6.4. Zeta Potential Measurement (ζ-Potential)

The electrochemical double layer (EDL) is a system used to explain the ζ-potential and consists of a system model in which a solid in contact with an aqueous solution assumes a surface charge; this charge provokes an interfacial charge distribution that is different from the charge distribution in the liquid phase. The EDL model compensates the surface charge and defines 2 layers: 1 stationary layer and 1 diffuse mobile layer. The ζ-potential is the potential at the boundary between the stationary and the mobile diffusive layers (shear plane).

This parameter can be measured experimentally through the gauge of the electrokinetic effect appropriate for solid samples. There are many different electrokinetic effects: electro-osmosis, electrophoresis, sedimentation potential, and streaming potential, and each of these effects needs the 2 phases to move relative to each other. This study used the streaming potential concept, which is generated by the tangential flow of liquid forced to pass through a granular powder (Figure 1a). The streaming potential is generated from the movement of a liquid in a capillary formed from the solid samples; in this situation the ions of the electrochemical double-layer are shared off their equilibrium position and shifted along the solid surface, resulting in a net charge separation leading to an electrical potential difference (Figure 1a). This so-called streaming potential is detected between electrodes located on both ends of the capillary.

ζ-potential measurement was performed using the SurPass 3 equipped with a cylindrical cell for granulate materials (Anton-Paar GmbH, Graz, Austria).

Measurements were performed using a cylindrical cell, and for each bone filler material, 1.5 g of granules were transferred in the cylindrical compartment of the cylindrical cell and mounted between the support disk and filters (with 25 µm mesh) on both sides of the granular sample plug (Figure 1a). A volume of 0.001 M KCl was used as the electrolyte solution. Three replicates for each test sample were analyzed. The analysis was performed according to the pH scan method. It consists of the measurement of the streaming potential at different pH points, between 9 and 5. The pH of the electrolyte solution was automatically modified by the instruments using 0.05 M HCl and 0.05 M KCl. At each pH point, 3 measurements were performed in order to condition the sample, then the 4th value was taken and reported.

To calculate the ζ-potential, the Helmholtz-Smoluchowski approximated equation was used:(2)$$\zeta =\frac{dU_{str}}{d\Delta p\;}\times \frac{\eta }{\varepsilon \times \varepsilon_{0}}\times \kappa_{B}$$
where *dU_str_*/*d*Δ*p* is the streaming potential coefficient, *κ**_B_* in the case of non-conducting samples is the electrical conductivity of the bulk electrolyte solution, and *η* and *ε* × *ε*_0_ are the viscosity and dielectric coefficient of the electrolyte solution. For diluted aqueous solutions, the viscosity and the dielectric coefficient of water were used. All calculations were performed by the instrument software.

### 2.7. In Vitro Release Study

The Pph release study was performed by using the Folin-Ciocalteu method to quantify the phenolic content in the release media and using the DPPH assay to quantify the antioxidant power; experimental details are described in the sections *Phenolic content of PRPE* and *Antioxidant power of PRPE*. The HPLC analysis was performed on the released solution according to the method described in section *UHPLC analysis*. Three specimens for each sample were immersed in ultrapure water (1 mL of dH_2_O for 0.2 g of SR) at 37 °C for 24 h, 72 h, and 1 week. At each time point, the release solution was removed and replaced with the same amount of fresh solution. A 0.2 g measure of Synergoss Red incorporates around 0.951 mg of polyphenols, which could be considered the initial amount of molecules.

### 2.8. In Vitro Biological Assessment

The biological tests were performed on SR at the concentration of 0.2 g/well, at direct contact. Cells were seeded directly on the top of the granulate. This approach was used for the cytotoxicity, inflammatory, and pro-osteogenic tests.

#### 2.8.1. Cytotoxicity Assay

The cytocompatibility test was performed using an osteoblast-like cell line. The SaOS2 osteoblast-like cell line was used in the cell adhesion experiments. The experimental cell culture medium (BIOCHROM KG, Berlin, Germany) consisted of Minimum Eagle’s Medium without L-glutamine, 10% fetal bovine serum, streptomycin (100 g/L), penicillin 100 U/mL, and 2 mmol/L L-glutamine in a 250 mL plastic culture flask (Corning^TM^ , Corning, NY, USA). Cells were cultured at 37 °C in a humidified incubator equilibrated with 5% CO_2_. Cells were harvested prior to confluence by means of a sterile trypsin-EDTA solution (0.5 g/L trypsin, 0.2 g/L EDTA in normal phosphate buffered saline, pH 7.4), re-suspended in the experimental cell culture medium, and diluted to 1 × 10^5^ cells per mL. The quantitative test was performed in direct contact. Briefly, SaOS2 cells were seeded on SR granulate (0.2 g/well SR). Quantitative cell viability was evaluated by MTT assay after 72 h. This assay evaluates the efficiency of the mitochondrial succinate dehydrogenase (SDH) enzyme of the citric acid cycle, making it possible to identify any toxic effect due to the reduction of enzyme activity. The MTT test requires, after microscopic examination, the cells to be incubated with 1 mg/mL solution of soluble tetrazolium salt (specifically, 3-(4.5-dimethylthiazol-2yl)-2.5 diphenyl tetrazolium bromide). During the subsequent 2 h of incubation at 37 °C, the succinate dehydrogenase enzyme caused the tetrazolium salts to be transformed first into a yellow, soluble substance and then into a blue water-insoluble product, the formazan. The greater the amount of precipitate, the higher the enzyme activity and, consequently, the number of metabolically active cells. We used as negative control the cells grown on the polystyrene plate without any stimulus, and as positive control the cells grown on polystyrene, with the addition of 20 µL of a solution of 0.08 mg/mL of sodium nitroprusside (NPS), as the apoptotic compound. The precipitate was dissolved with dimethyl sulfoxide and spectrophotometrically measured at a wavelength of 570 nm, providing the optical density (OD) to be used to calculate cell viability. Results are presented as the mean ± standard deviation (SD) of 3 biological samples (*n* = 3).

#### 2.8.2. Inflammatory and Oxidative Stress Response

The inflammatory response test was conducted by using the direct contact approach between cells and the SR material, at the concentration of 0.2 g/well. The murine macrophage cell line J774a.1 (European Collection of Cell Cultures) was maintained in Dulbecco’s modified eagle medium (Gibco Invitrogen, Cergy-Pontoise, France) supplemented with 10% fetal bovine serum, penicillin (100 U mL^−1^), streptomycin (100 μg mL^−1^), and 4 mM l-glutamine. Cells were grown in a 100% humidified incubator at 37 °C with 10% CO_2_ and passaged 2–3 days before use. J774a.1 cells (1.05 × 10^6^ mL^−1^) were seeded onto 12-well tissue culture polystyrene plates (Falcon^®^ Corning^TM^, Corning, NY, USA), containing the SR granulate in direct contact. After 4 h, the RNA from J774.a1 cells was isolated by using the Maxwell^®^ RSC simplyRNA Cells Kit (Promega Corporation, Madison, WI, USA), by following the manufacturer’s instructions. RNA quantitation was performed using the Quantifluor RNA system kit in the Quantus Fluorometer (both from Promega Corporation), and the obtained total RNA was reverse-transcribed using a High-Capacity cDNA Reverse Transcription Kit in the Thermal Cycler 2720 (both from Thermo Fisher Scientific, Waltham, MA, USA) at the following conditions: 10 min at 25 °C, 120 min at 37 °C, and 5 min at 85 °C, and maintained at 4 °C until further experimentation. RT-qPCR was performed following the Fast running protocol of the TaqMan^®^ FastAdvanced Master Mix (Thermo Fisher Scientific, Inc.) in the QuantStudio 5 Real Time PCR System (Thermo Fisher Scientific, Inc.) using designed murine TaqMan^®^ assays (Thermo Fisher Scientific, Inc.) to quantify gene expression of the following genes: interleukin-1β (IL-1β; ID: Mm01336189_m1), interleukin-6 (IL-6; ID: Mm99999064_m1), tumor necrosis factor alpha (TNF-α; ID: Mm00443258_m1), interleukin 10 (IL-10; ID: Mm99999062_m1), inducible nitric oxide synthase (iNOS/NOS2; ID: Mm00440502_m1), catalase (CAT; ID: Mm00437992_m1), glyceraldehyde 3-phosphate dehydrogenase (GAPDH; ID: Mm99999915_g1), and tyrosine 3-monooxygenase/tryptophan 5-monooxygenase activation protein ζ (YWHAZ; ID: Mm03950126_s1).

Real-time PCR was performed in technical duplicate for all samples and targets in a total volume of 25 µL and the amplification was performed as follows: hold at 50 °C for 2 min, hold at 95 °C for 2 min, 40 cycles at 95 °C for 1 s and 60 °C for 20 s. All transcripts were normalized according to the comparative threshold cycle (ΔΔCt) method [42], consisting of the normalization of the number of target gene copies versus the endogenous reference genes YWHAZ and GAPDH. The Ct is defined as the fractional cycle number at which the fluorescence generated by cleavage of the probe passes a fixed threshold baseline, when amplification of the PCR product is first detected. Subsequently, the evaluation of expression of target genes relative to the reference untreated sample group polystyrene was performed. Results are presented as the mean ± SD of 3 biological samples (*n* = 3).

#### 2.8.3. Pro-Osteogenic Response

Human osteoblast-like SAOS2 cells were used in cell growth experiments. Experimental cell culture medium (BIOCHROM KG, Berlin) consisted of Minimum Eagle’s Medium without l-glutamine, 15% fetal bovine serum, streptomycin (100 μg/L), penicillin (100 U/mL), and 2 mM l-glutamine in 250 mL plastic culture flasks (Corning™, Corning, NY, USA). Cells were cultured without the addition of any osteogenic differentiation media at 37 °C in a humidified incubator equilibrated with 5% CO_2_. The cell suspension was obtained by adding 2 mL of a sterile 0.5% Trypsin-EDTA solution (ref 15400-054 - GIBCO by Thermo Fisher Scientific, Waltham, MA, USA), to a 250 mL cell culture flask, re-suspended in the experimental cell culture medium, and diluted to 8.5 × 10^4^ cells/mL. Cells were seeded at the density of 8.5 × 10^4^ cells/mL onto 12-well tissue culture polystyrene plates (Falcon^®^ Corning^TM^, Corning, NY - USA) containing the Synergoss samples at the amount of 0.2 g/well. After 72 h, 5 days, and 7 days, RNA from SaOS2 cells was isolated, quantitated, and reverse-transcribed as described in the section Inflammatory and oxidative stress response.

The expression of YWHAZ, GAPDH, ALPL, COL1a1, SPARC, RANKL, OPG, and MMP9 genes as osteoblast differentiation markers was assessed by RT-qPCR using designed human TaqMan^®^ assays (Thermo Fisher Scientific, Inc.): YWHAZ (ID: Hs03044281_g1), GAPDH (ID: Hs00266705_g1), collagen type I alpha 1 chain (COL1a1; ID: Hs00164004_m1), receptor activator of nuclear factor kappa-Β ligand (TNFSF11/RANKL; ID: Hs00243519_m1), osteoprotegerin (TNFSFR11b/OPG; ID: Hs00900358_m1), alkaline phosphatase (ALPL; ID: Hs01029144_m1), osteonectin (SPARC; ID: Hs00234160_m1), and matrix metalloproteinase 9 (MMP9; ID: Hs00234579_m1). Real-time PCR was performed in technical duplicate for all samples and targets in a total volume of 25 µL, and the amplification was performed as described in the section Inflammatory and oxidative stress response. All transcripts were normalized according to the comparative threshold cycle (ΔΔCt) method, consisting of the normalization of the number of target gene copies versus the endogenous reference genes YWHAZ and GAPDH. Subsequently, an evaluation of the expression of target genes relative to the reference untreated sample group polystyrene was performed. Results are presented as the mean ± SD of 3 biological samples (*n* = 3).

### 2.9. Statistical Analysis

The statistical analysis was performed using both the PAST [43] free software (Release 3.18, Professor Øyvind Hammer, Natural History Museum, University of Oslo, Oslo, Norway) and the Real Statistics Resource Pack software (Release 6.6.2, Copyright (2013–2021), www.real-statistics.com, Charles Zaiontz, Milano, Italy) [44].

Gene expression and cell viability data were tested for differences among group means. Data were first tested for normality, with the Shapiro-Wilck test and for homogeneity of variances, with the Levène’s test. When both assumptions were satisfied, one-way or two-way analysis of variance (ANOVA), followed by the Tukey’s honestly significant difference (HSD) post-hoc test, was applied. In the cases with unequal variances, we applied the Welch’s ANOVA followed by the Tukey’s HSD post-hoc test. Results were considered statistically significant at *p* < 0.05 (* = *p* < 0.05; ** = *p* < 0.01; *** = *p* < 0.001 and **** = *p* < 0.0001).

## 3. Results

### 3.1. Croatina PRPE Characterization

Grape pomace was collected and stored at −20 °C under vacuum until the beginning of the extraction process. Prior to functionalization of the ceramic bone filler, PRPE was characterized by different techniques, and the polyphenolic pattern was identified and quantified.

Analysis of the phenolic content of PRPE from the grape variety Croatina showed an initial amount of 3.17 mg/mL of GAE. The free radical scavenging capacity of the extract was also evaluated by DPPH assay, which is considered a valid, accurate, easy, and economical method to evaluate the radical scavenging activity of antioxidants, because the radical compound is stable and does not need to be generated. Croatina PRPE reduced radicals by 72.8%, which suggests a high antioxidant power of the polyphenolic molecules contained in the extract.

### 3.2. UHPLC Analysis

The polyphenolic pattern of PRPE obtained by HPLC-DAD analysis allowed classification into different classes of the separated peaks in the chromatograms: phenolic acids and flavonoids (maximum absorbance at 277–280 nm), hydroxycinnamic acid (maximum absorbance 313–330 nm with sometimes a shoulder of 290 nm), flavonols (maximum absorbance at 350–385 nm), and anthocyanidines (maximum absorbance at 280–320 nm with a specific absorbance at 525 nm).

By using available standard solutions, it was possible to identify and quantify several specific polyphenols that characterize and define the fingerprint of the analyzed PRPE: gallic acid, rutin, quercetin, tannic acid, catechin, epicatechin, malvidin-3-glucoside, procyanidin B2, myricetin, quercitrin, kaempferol, isorhamnetin, quercetin-3-glucuronide, trans p-coumaroyl tartaric acid, and epigallocatechin gallate.

This method made it possible to identify and quantify just a percentage of the total amount of polyphenolic molecules, in particular a value equal to 1.808 mg/mL, which is around 57% of the total polyphenolic content of PRPE (3.17 mg/mL), calculated with the Folin-Ciocalteu method.

Several peaks were not comparable with standard molecules. In fact, in nature polyphenols usually occur conjugated to sugar and organic acids. Furthermore, not all naturally occurring molecules are present as standard compounds.

However, the obtained results showed that PRPE from Croatina are characterized by a significant amount of phenolic acids (tannins and gallic acid), flavonoids (catechin, procyanidin B2, and epicatechin), and flavonols (quercetin, myricetin, quercetin-3-glucuronide, quercitrin, kaempferol, and isorhamnetin).

These results are crucial, because although the polyphenolic molecules were not identified in their entirety, they demonstrated which molecules are released more in the early stages and which ones remain incorporated in the granules, thus conferring an antioxidant power to the bone filler itself.

### 3.3. Surface Analysis

#### 3.3.1. Morphological Characterization

Morphology and surface area play an important role in the interaction of bone cells with biomaterial surfaces. Micro-/macrosurface pores and roughness promote mechanical interlocking and bone ingrowth, influence the tissue response, and increase the fixation and stability of the implant. Figure 2a,b show SEM images of the surface of granules of the bone filler control, whereas Figure 2c,d show Synergoss Red, at different magnifications (a–c: 1000×; b–d: 15,000×).

The production process for both materials allows the formation of a rough surface, characterized by a diffused microporosity that influences the surface area and consequently the osteogenic activity of the bone filler [45]. Roughness and porosity provide support to bone cells and contribute to the stimulation of cell proliferation and differentiation [45,46,47]. The presence of antioxidant molecules on granule surfaces was investigated by ATR-IR, XPS, and ζ-potential analysis.

#### 3.3.2. ATR-IR

Both the untreated (BF) and the PRPE-treated (SR) ceramic bone fillers were analyzed in order to provide information related to the chemical composition and main functional groups of the material. In Figure 3a, the spectra of the bone fillers with and without PRPE functionalization are reported (from 500 cm^−1^ and 4000 cm^−1^). Figure 3b shows a focus on the peaks related to the inorganic phase (1500–500 cm^−1^), with a clear calcium phosphate composition of Synergoss Red. Both the bone fillers showed typical bands associated with the triply degenerated asymmetric stretching mode of P-O bonds of phosphate groups in the range between 950 and 1140 cm^−1^ [3]. The peak at 1125 cm^−1^ is due to tricalcium phosphate, while peaks at 1025 cm^−1^ and 1010 cm^−1^ are typically associated with HA, overlapped with the component due to β-TCP [3,48]. Interesting results were obtained comparing the Synergoss Red material with the bone filler control and PRPE from lyophilized Croatina extracts used for the treatment of the bone filler (Figure 3c).

In particular, the spectrum of Synergoss Red between 1450 cm^−1^ and 1800 cm^−1^ showed an absorption similar to the one obtained for the PRPE. Bands at 1645 cm^−1^, which are clearly present in Synergoss Red and PRPE, could be related to the C=C stretching vibration of aromatic rings; furthermore, some evidence in the literature associates these bands with the flavonoids, whereas the bands between 1490 cm^−1^ and 1600 cm^−1^ were probably due to the presence of the aromatic -C=C- bond [49,50].

#### 3.3.3. XPS

Through XPS analysis it is possible to assess the surface chemical composition of materials. The chemical composition of the tested Synergoss Red surface was compared with that of the bone filler control, composed of the same ceramic material without PRPE functionalization. Spectra of the analyzed bone filler samples were dominated by the peaks of oxygen, carbon, calcium, and phosphorus. Interesting results were obtained by analyzing the surface stoichiometry reported in Table 3.

The surface composition of Synergoss Red showed significantly more carbon and significantly less calcium and phosphorus compared to the bone filler control. The increase of the carbon contribution to the surface stoichiometry and the attenuation of the signal from the ceramic “core” elements Ca and P, was in agreement with the presence of a surface layer of organic molecules, confirming the results obtained by the ATR-IR analysis [3,51,52].

#### 3.3.4. ζ-Potential Measurement

This parameter is an indirect measure of the surface charge and therefore, of the chemical groups present on a surface, and is obtained by flowing an aqueous solution with known ionic strength in the capillary channel through the ceramic granules. In Figure 1b, the pH scan curves for the material functionalized with PRPE (Synergoss Red) and for the control material without polyphenols (bone filler control) are shown.

First, it must be noted that no isoelectric point was detected, and that both samples showed a negative value in the whole tested pH range. Hence, in every case, the negative charges dominated the aqueous interface. Calcium phosphate materials normally show a negative charge on the surface and acidic behavior [53]. However, the introduction of polyphenolic molecules on the surface made the ζ-potential even more negative and acidic. Furthermore, the ζ-potential curve for Synergoss Red showed a value almost constant in all pH ranges; this means that homogeneous chemical functions with acidic behavior were present on the surface, determining the charge and therefore the surface ζ-potential [54]. The formation of a plateau during measurement revealed the presence of a single value of pKa for the surface, which means that the interfacial charge behavior was dominated by chemical functionalities present on the surface rather than by ions derived from the solution [3].

### 3.4. In Vitro Release Study

In order to understand the amounts and types of molecules that are released in situ by Synergoss Red, an in vitro release study was conducted for 1 week. At each time point (24 h, 72 h, and 1 week), FC, DPPH and UHPLC tests were performed. Results are shown in Table 4, which reports the phenolic content and antioxidant power of the released solution at each time point, and in Table 5, which reports part of the phenolic composition of the released solution.

The results showed that after 24 h, the material released the highest amount of polyphenols, namely, 0.55 mg/mL GAE, with a measured antioxidant power of 8.7%. These values decreased over time, and after 1 week, the observed phenolic content was 0.04 mg/mL GAE, and the antioxidant power, 0.5%. After 1 week, around 75% of total phenolic content incorporated in the granules was released (0.7226 mg out of 0.951 mg).

The UHPLC analysis performed on the released solution allowed us to identify and quantify some of the released phenolic molecules (Figure 4a–c).

Table 5 reports the amount of released polyphenols, compared with the amount of polyphenols in the PRPE used for the treatment of the ceramic bone filler. The percentage of each molecule released was calculated in comparison with the initial amount in the PRPE extract used for the treatment of ceramic bone filler (Table 5).

After 1 week of release, the surface charge was still dominated by the presence of certain specific polyphenolic molecules. Water-soluble polyphenols, such as gallic acid, catechin, epicatechin (EC), procyanidins, and EPGCG, were released in higher percentages compared with flavonols such as kaempferol, quercetin, and isorhamnetin, which were characterized by reduced solubility in the aqueous environment.

We hypothesized that the molecules still present on the surface of Synergoss Red after 1 week generated an antioxidant potential. For this reason, a DPPH test on the Synergoss Red material itself was conducted at the end of the release week. The results confirmed our hypothesis because the measured antioxidant power was 31.80 ± 0.68%. Furthermore, the presence of polyphenols on the ceramic granules after 1 week of release in aqueous solution was also assessed through an analysis of the ζ-potential generated at the material-aqueous interface by the surface charge of the tested material as a function of pH conducted on Synergoss Red after 1 week of release (Figure 5a). Represented data show negative values over the entire tested pH range at both time points and a slight difference between the two samples; in particular, the results for the material after 1 week were more negative and acidic compared with the material at the beginning of the release study (Figure 5b).

### 3.5. In Vitro Biological Assessment

#### 3.5.1. Cell Viability Assay

The cytocompatibility of the tested biomaterials was assessed in vitro by MTT assay. We found that both the bone filler control and Synergoss Red at the concentration of 0.2 g/well did not exert any significant cytotoxic effect on SaOS2 osteoblast-like cells, thus confirming that these materials are not cytotoxic. As shown in Figure 6a, the bone filler control showed a percentage of cell viability comparable to that observed in the negative control (negative control: 99.5 ± 2.1%; bone filler control: 99.4 ± 3.3%), whereas only a slight downregulation of cell viability (95.1 ± 2.6%) was observed in cells incubated with Synergoss Red, thus indicating excellent cytocompatibility according to the international standard ISO 10993-5:2009. The released polyphenols did not induce any cytotoxic effect, and this was also confirmed by the optical images (Figure 6b), which showed normal morphology of cells after 72 h in contact with Synergoss Red.

#### 3.5.2. Inflammatory and Oxidative Stress Response

In order to investigate the inflammatory reaction of macrophage cells to incubation in direct contact with Synergoss Red, expression of the pro-inflammatory genes IL-1β, IL-6, and TNF-α was evaluated (Figure 7a). Compared to the untreated control, polystyrene, the bone filler control showed significantly higher expression of all the examined genes (IL-1β: 14.6-fold; IL-6: 4.6-fold; TNF-α: 3.4-fold, *p* < 0.0001 for all), statistically significant also relatively to Synergoss Red (IL-1β and IL-6: *p* < 0.0001; TNF-α: *p* < 0.05). It is interesting to note that the presence of Croatina PRPE in Synergoss Red induced in macrophages a lower pro-inflammatory gene expression profile compared to the bone filler control, with values slightly higher than those found in polystyrene (IL-1β: 1.2-fold, not significant; IL-6: 2.4-fold, *p* < 0.001; TNF-α: 3-fold, *p* < 0.0001). In addition to pro-inflammatory genes, the anti-inflammatory IL-10 gene was also evaluated. Compared to polystyrene, mRNA levels for IL-10 in cells incubated with Synergoss Red were 3.1-fold higher (although not significant), whereas no changes in IL-10 expression were observed in cells incubated with the bone filler control (Figure 7b). As polyphenols can act both as anti-oxidant and pro-oxidant agents [55], the expression of genes involved in the oxidative stress response, such as iNOS and CAT, was also investigated (Figure 7b).

Specifically, iNOS mRNA levels were higher in cells incubated with both the bone filler control (2.6-fold, *p* < 0.0001) and Synergoss Red (1.4-fold, *p* < 0.05) than with polystyrene, whereas it is evident that polyphenols in Synergoss Red significantly reduced the extent of iNOS expression compared to the bone filler control (*p* < 0.0001). No changes were observed for CAT expression in the ceramic bone filler samples compared to polystyrene.

#### 3.5.3. Osteogenic Response

Cells were investigated for a pro-osteogenic response to incubation in direct contact with the bone filler control and Synergoss Red. Analysis of Col1a1 showed a time-dependent increase in the mRNA levels of cells with all treatments (Figure 8b). In particular, the bone filler control and Synergoss Red induced an opposite Col1a1 expression at all three time points in the analysis; in fact, cells incubated with the bone filler control showed decreased levels compared to incubation with polystyrene (72 h: 0.6-fold; 5 d: 0.7-fold; 7 d: 0.9-fold) and Synergoss Red, whereas Synergoss Red induced significantly higher levels relative to both polystyrene (72 h: 1.2-fold; 5 d: 1.8-fold, *p* < 0.01; 7 d: 2.6-fold, *p* < 0.0001) and the bone filler control at all three time points (72 h: *p* < 0.05; 5 d and 7 d: *p* < 0.0001). In addition, the observed increase in Synergoss Red samples was significantly time-dependent.

Analysis of SPARC gene expression showed an overall decrease in cells incubated with the bone filler control as compared to both polystyrene (72 h: 0.8-fold; 5 d: 1.2-fold; 7 d: 2-fold) and Synergoss Red, particularly evident and statistically significant at the 7 d time point (*p* < 0.0001) (Figure 8c). Concerning ALPL expression, a time-dependent increase was observed for all the analyzed samples, with the bone filler control samples showing similar values to polystyrene (72 h: 0.96-fold; 5 d: 1.95-fold; 7 d: 2.6-fold), whereas Synergoss Red samples showed reduced levels (72 h: 0.7-fold; 5 d: 1.6-fold; 7 d: 1.9-fold), significant at 7 d incubation time (*p* < 0.0001 as compared to polystyrene and *p* < 0.01 as compared to the bone filler control) (Figure 8a).

Results of RANKL cytokine analysis showed again a substantial time-dependent increase in mRNA levels in untreated cells, whereas the bone filler control and Synergoss Red significantly reduced RANKL expression at the three time points (bone filler control: 72 h: 0.6-fold, *p* < 0.05; 5 d: 0.8-fold, *p* < 0.01; 7 d: 0.7-fold, *p* < 0.0001—Synergoss Red: 72 h: 0.1-fold, *p* < 0.0001; 5 d: 0.3-fold, *p* < 0.0001; 7 d: 0.3-fold, *p* < 0.0001) (Figure 9a). It is interesting to note that cells incubated with Synergoss Red showed the lowest values already at 72 h incubation. By comparing expression of RANKL with that of its decoy receptor OPG (Figure 9b), it is evident that cells incubated with the bone filler control or Synergoss Red behave in a different manner; in fact, the bone filler control induced a progressive increase in OPG transcript levels, as compared to polystyrene at 72 h (72 h: 1-fold; 5 d: 2-fold; 7 d: 3-fold), but similar to OPG expression levels in polystyrene at the same time point, whereas Synergoss Red induced an early significant increase at 72 h (*p* < 0.01), with a further downregulation at 5 and 7 d (72 h: 2.1-fold; 5 d: 1.1-fold; 7 d: 1.3-fold). Analysis of RANKL/OPG ratio showed the ability of Synergoss Red to dramatically downregulate the net effect of RANKL, especially at 72 h (Figure 9c); in fact, the value of the ratio was significantly lower (0.07-fold, *p* < 0.0001) than that of both polystyrene and the bone filler control (0.6-fold, *p* < 0.0001). At 5 and 7 d time points, cells of the untreated samples also showed a decrease in the ratio (5 d: 0.4-fold; 7 d: 0.6-fold), as compared to the control at 72 h, although higher than in the bone filler control (5 d: 0.37-fold; 7 d: 0.2-fold, *p* < 0.0001) and in Synergoss Red (5 and 7 d: 0.2-fold, *p* < 0.05 for 5 d and *p* < 0.0001 for 7 d).

Analysis of MMP-9 showed the bone filler control and Synergoss Red to have similar effects on its expression; in fact, both ceramics downregulated MMP-9 mRNA levels at 5 and 7 d (bone filler control: 72 h: 0.7-fold; 5 d: 0.9-fold; 7 d: 0.5-fold, *p* < 0.01—Synergoss Red: 1.1-fold; 0.7-fold, *p* < 0.05; 7 d: 0.8-fold, *p* < 0.05) compared to polystyrene (Figure 8d).

## 4. Discussion

Periodontal and peri-implant regeneration has aroused great interest [1,56] in bone regeneration. In the challenging cases of peri-implant infection and inflammation, the available ceramic materials are unable to control oxidative stress, which could degenerate in a complex peri-implantitis.

Polyphenols have been widely recognized as molecules with health benefits; their anti-oxidant and anti-inflammatory properties make them interesting molecules for the treatment of peri-implantitis [57]. Several studies have investigated their effect on bone regeneration and in particular, their effect on osteoblast differentiation, bone mass regeneration, and inhibition of osteoclastogenesis and bone loss [58,59]. Accordingly, with respect to the clinical need for new materials able to control bone regeneration during peri-implant inflammation, in this work we investigated the effects of a readily available polyphenol source, namely, a pomace extract, on the clinically relevant properties of a granulate biphasic bone filler.

The polyphenol mix of PRPE from Croatina grape is characterized by a great variety of different amounts of polyphenols (3.17 mg/mL), which worked together to exert an antioxidant power of 72.83%. The analysis of the phenolic composition of the extract from Croatina grape suggests that flavanols have the highest concentration in the extract, followed by the flavonols such as quercetin, myricetin, kaempferol, and isorhamnetin, which are cited in several studies as polyphenols involved in many bone regeneration mechanisms [60,61,62]. However, the mixture of different polyphenolic compounds, specific for Croatina grape and replicable from different batches of the same grape pomace mass, makes these extracts extremely interesting, since it is possible to take advantage of the potential of a large number of polyphenolic classes. Different studies suggest the efficacy of the use of polyphenols to treat dental diseases [57,63,64], but the normal route of administration is via oral intake, which is subject to different variables that could reduce or at least make unpredictable the outcomes.

In order to maximize the potential effect of these molecules, it is important to ensure a local release [65]. However, in the case of a periodontal defect, the use of a bone filler that can exert both osteoinductive and osteoconductive potential is still important [66,67]. In our research, the first step was to functionalize the ceramic granulates with PRPE and demonstrate the presence of polyphenols on the surface of the material. ATR-IR analysis clearly showed the presence of a peak related to the aromatic rings, which are characteristic of phenols. Particularly intense were the peaks between 1500 cm^−1^ and 1800 cm^−1^, indicating the high quantity of grafted polyphenols on the analyzed surface. Specifically, the IR spectra of Synergoss Red showed a shape similar to that identified in the spectra of PRPE at 1700 cm^−1^ due to the COOH stretching vibration, at 1715 cm^−1^ due to the aromatic ring, and at 1645 cm^−1^ and 1637 cm^−1^, which correspond to the C=C stretching vibration of the aromatic ring and to C=O stretching vibrations, respectively. Furthermore, XPS analysis confirmed these results; in fact, a reduction of Ca and P% amount and an increase of the C percentage, have been evaluated.

The ζ-potential analysis, together with the results obtained with the other surface analysis techniques, further proved the presence of polyphenols on the surface of granulates. Biphasic calcium phosphate ceramics (HA and βTCP) show strong reactivity and chemical instability at low pH, and this fact may influence the measurements; therefore, the ζ-potential analysis was conducted at a pH range of 8.5–5.5. Furthermore, the chosen pH range was more interesting from a clinical point of view. In fact, as reported by Nyako et al., the pH close to a failed implant is more alkaline (around pH 7.2) compared to the pH measured in the peri-implant crevicular fluid found around a successful implant (around pH 6.9) [68]. Mombelli et al. stated that gram-negative anaerobic bacteria play a crucial role in peri-implant infections and grow in alkaline pH environments [69]. The Synergoss Red material was compared with the bare ceramic granulate, namely, the bone filler control, in order to emphasize the contribution of the substrate of polyphenols. Synergoss Red showed a plateau between pH 6 and 8. The formation of a plateau during a zeta potential measurement and a single value of surface pKa mean that the interfacial charge behavior is dominated by chemical functionalities on the surface of the material rather than by ions derived from the solution [3]. Calcium phosphate materials usually show a negative charge on the surface and acidic behavior due to the dissociation of ceramic compounds in phosphate ions PO_4_^3−^, which bind with H_3_O^+^ in solution to form acidic species [53]. However, the dissociation of -OH groups of the carboxyl group and of phenolic -OH groups present in the molecular structure results in a much more negative surface ζ-potential for Synergoss Red compared to the bone filler control. PRPE showed a high amount of phenolic acids, tannins, flavanols, and flavonols, which are characterized by multiple phenolic hydroxyl and carboxylic groups and dissociate in the analyzed range (pKa = 4.44 and pKa = 8, respectively). Furthermore, Synergoss Red showed a more stable curve also at acidic pH, thus denoting a continuous layer of grafted biomolecules on the surface of the functionalized samples and the shielding ability of grafted polyphenols with respect to surface reactivity of the calcium phosphate bone graft material. On the other hand, an increase in the slope of the curve was noted for the bone filler control at acidic pH, which could be associated with the chemical instability of the material when the pH decreases. This consideration could suggest that the layer of polyphenols could protect the material from degradation at low pH.

Many polyphenols have been reported to interact with cells inducing specific molecular pathways [33]. For this reason, it is important to understand which kinds of polyphenols could be released and how much antioxidant power is exerted by those molecules. High performance liquid chromatography allows an understanding of the distribution of the polyphenolic classes and the amounts of specific molecules. After 1 week, it was possible to speculate that the flavan-3-ol molecules and, in particular, catechin, EPC, EPGCG, procyanidin B2, and phenolic acids such as gallic acid were released in the highest amounts. Conversely, flavonoids such as quercetin, myricetin, kaempferol, rutin, isorhamnetin, quercitrin, and quercetin-3-glucuronide were still present on the surface of Synergoss Red. This was due to differences in the solubility of the molecules in an aqueous environment, which allowed a selective and sustained release over time. Another interesting aspect is that, after 1 week of release, the antioxidant power of the material itself was still 31.80 ± 0.68%, which signifies prolonged antioxidant power of the material subsequent to the early release of soluble molecules from the surface. This aspect makes the material effective for a long time against infection-derived oxidative stress. The presence of a polyphenolic layer on the surface of the calcium phosphate Synergoss Red was confirmed by ζ-potential analysis of the granulates after 1 week of release in an aqueous environment. The surface ζ-potential of Synergoss Red after 1 week of release was, as expected, slightly less negative and acidic than Synergoss Red as prepared at time 0, because a percentage of the initial polyphenols was released. However, the presence of flavonol molecules on the surface of granulates still made the surface ζ-potential of Synergoss Red more negative than the material without polyphenols (bone filler control) and confirmed the presence of an antioxidant layer for a prolonged time.

The two tested biomaterials were shown to be cytocompatible and to not induce any cytotoxic effect; furthermore, the extent of inflammation induced by Synergoss Red was notably low, with higher IL-10 expression. This can be explained by the anti-inflammatory properties of polyphenols that were shown to also promote the M2 anti-inflammatory macrophage state [70], characterized by decreased pro-inflammatory cytokines and iNOS and elevated IL-10 levels. An extensive literature shows polyphenols to be able to reduce pro-inflammatory cytokines [71] and, in particular, to modulate M1-M2 polarization and inhibit inflammatory mediators, including quercetin [72], epigallocatechin-3-gallate [73], gallic acid [74], tannic acid [75], resveratrol [76], procyanidin b2 [77], myricetin [78], apigenin [79], and luteolin [80]. Macrophage polarization toward a higher M1/M2 ratio was shown to be involved in the progression of gingivitis to periodontitis [81], with sustained IL-1β, IL-6, and TNF-α and diminished IL-10 expression [82]. Additionally, polyphenols such as quercetin and EPGCG were shown to be particularly effective in mitigating the inflammatory response and oxidative stress in periodontal disease [83].

From our RT-qPCR results, it’s clear that the early pro-inflammatory response induced by the biomaterial itself was mitigated by the polyphenolic extract contained in Synergoss Red, thus suggesting that polyphenols are able to direct the early inflammatory process towards a less inflammatory and reparative late phase, and that their use could be tailored in order to prevent any uncontrolled or disproportionate response that may occur.

This possible role of polyphenols in promoting faster tissue regeneration was also investigated in the field of bone regeneration, given their proven direct and indirect protective actions on bone, such as decreased bone loss thanks to their anti-inflammatory and antioxidant properties, reduced osteoclastogenesis, and enhanced osteoblastogenesis. We found diminished expression of ALPL, which is primarily involved in the early mineralization process of bone matrix [84], and increased mRNA levels of collagen type 1 and osteonectin in osteoblast cells incubated with Synergoss Red. Given that the major expression of collagen type 1 and osteonectin is found during the early phases of bone formation and specifically before the mineralization step [85], it is evident that polyphenols of Synergoss Red induced osteoblast cells to retain a more immature state, thus tending to produce more extracellular bone matrix. Polyphenols from wine extracts were shown to induce higher expression of Col1a1 and SPARC also in human mesenchymal stem cells [36], and specifically, quercetin [86], kaempferol [87], and catechins [88] were shown to direct osteogenic differentiation in different mesenchymal stem cell types. In addition, the enhanced Col1a1 expression observed in cells incubated with Synergoss Red could be of particular importance in treating periodontal disease, given that expression of type I collagen mRNA is shown to be reduced in the proximity of periodontal pockets of patients with periodontitis [89].

Thanks to their anti-inflammatory action, which was demonstrated to be the main osteoanabolic effector, polyphenols were shown to induce multiple anabolic effects on bone through targeting several molecular signaling pathways both directly and indirectly [33]. In particular, the combination of their anti-inflammatory and antioxidant properties makes them effective functional compounds to be exploited in the prevention and treatment of different pathological conditions. The displayed antioxidant potential of the polyphenolic extract contained in Synergoss Red is a result of the presence of different molecules characterized by strong antioxidant power [90]; first and foremost, gallic acid and tannic acid in addition to quercetin, catechins, EPGCG, and myricetin contribute to the observed bone protective effects. By scavenging ROS, antioxidant molecules are able to mitigate bone loss deriving from ROS-induced apoptosis of osteoblasts and from an imbalanced bone remodeling process [91]. High levels of ROS are implicated in the reduction of osteoblast survival, differentiation, and activity and in enhancement of the RANKL/OPG ratio [92]. RANKL binds to the RANK receptor [93], which is shown to be inhibited by most polyphenolic compounds. For example, kaempferol is able to attenuate in osteoblasts the TNF-induced promotion of pro-inflammatory cytokine expression [94]. Therefore, polyphenols that interfere with the TNF receptor signal, such as p-hydroxycinnamic acid [95], epigallocatechin-3-gallate [96], quercetin [88], and catechins [97], indirectly contribute to mitigate TNF-α inhibition of the TGF-β/SMAD and Wnt/β-catenin signals involved in osteogenic development [98].

In our study, RANKL mRNA levels were significantly lowered by both the bone filler control and Synergoss Red; analysis of the RANKL/OPG ratio clearly demonstrated that the two biomaterials act on this pathway through two different mechanisms. The first induces expression of OPG at late time points (starting from 5 d incubation) with a parallel slight downregulation of RANKL, whereas Synergoss Red, thanks to its phenolic composition with well-known anti-inflammatory actions, already dramatically affects RANKL expression levels by the 72 h incubation time point and continues to hold them at very low values also at the 5 and 7 d time points, with the sole upregulation of OPG at 72 h. These observations correlate well with the anti-inflammatory results, which showed lowered expression of pro-inflammatory cytokines in addition to iNOS and MMP-9 following incubation with Synergoss Red. Modulation of the RANKL/OPG ratio was shown to be induced by several polyphenols [33], among them EPGCG, quercetin, rutin, and catechins, thus demonstrating the high potential of polyphenols in the prevention of bone resorption. Furthermore, decreased iNOS expression determines a lower NO radical production and, as such, contributes to maintaining a lower oxidant environment. To this end, polyphenols such as quercetin, quercitrin [99], and catechins [100] were already shown to be effective.

Matrix metalloproteinases (MMPs) are involved in extracellular matrix remodeling and, as such, are required to be tightly regulated in order to avoid any uncontrolled degradation process, which is found to be behind all the pathological conditions involving tissue destruction [101]. Polyphenols [102] and in particular, EPGCG, catechins, and also proanthocyanidins were shown to be particularly effective in downregulating MMP-9 expression induced by bacteria involved in the pathogenesis of periodontal disease, such as *P. gingivalis*. In our study, Synergoss Red induced a significant decrease in MMP-9 expression at late incubation stages. The ability of Synergoss Red to downregulate MMP-9 is notable; in fact, MMPs and in particular, MMP-9, are implicated in periodontal tissue destruction in chronic periodontitis [103].

The results obtained in this study are promising in the interest of directing a bone regeneration process, in which a controlled inflammatory response and an antioxidant biological environment are both necessary, particularly in the case of oral diseases, in which a dysregulated inflammatory response is also consequent to the presence of different periodontal pathogens. However, given the ability of Synergoss Red to both provide control of inflammation and oxidative stress and enhance the deposition of early bone matrix in osteoblast cells, its use could be indicated also in other bone applications.

Ongoing research aimed at evaluating the influence on inflamed tissues and, more generally, on bone repair, will further contribute to better elucidation of the effects exerted by Synergoss Red also in vivo, thus providing more detailed and substantial data with applications in the oral context.

## 5. Conclusions

In conclusion, our results show that Synergoss Red is a cytocompatible biomaterial that induces expression of the main genes involved in early bone matrix deposition, downregulates inflammation, and modulates osteoclastogenesis due to the presence of polyphenol-rich pomace extracts that possess anti-inflammatory and antioxidant properties.

## Figures and Tables

**Figure 1 jfb-12-00031-f001:**
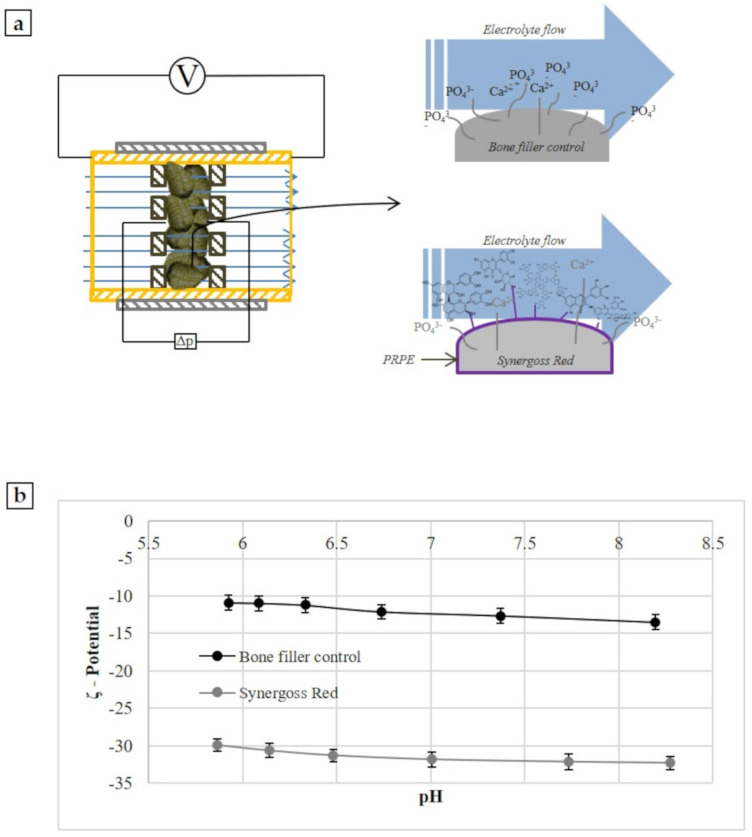
**Figure 1**. ζ-Potential measurements of Synergoss Red and Bone filler Control. (**a**) Graphical images of the principle of ζ-potential measurement for solid surfaces; (**b**) pH scan curve from 8.5 to 5.5.

**Figure 2 jfb-12-00031-f002:**
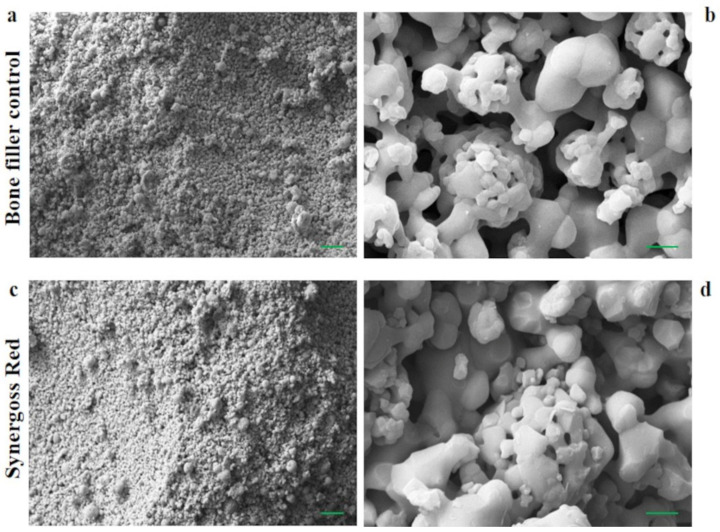
Images of the morphological structure obtained by scanning electron microscopy of (**a**–**c**) Synergoss Red bone filler at 1000× magnification, scale bar 2 μm and (**b**–**d**) 15,000× magnification, scale bar 20 μm.

**Figure 3 jfb-12-00031-f003:**
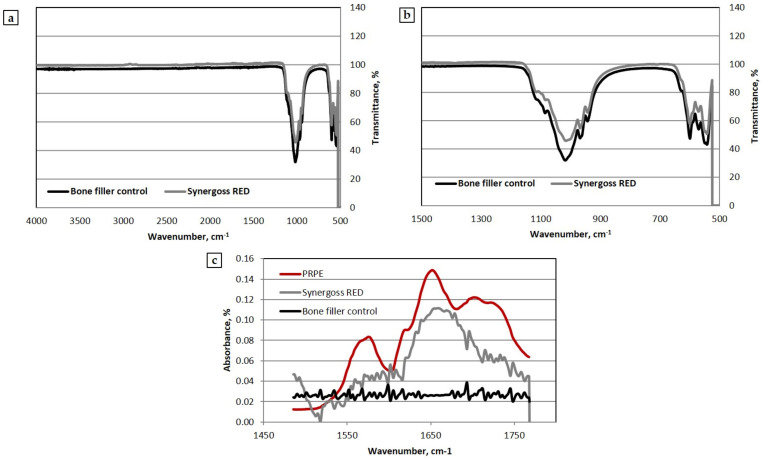
ATR-IR spectra of the analyzed bone fillers (SR and BF) and lyophilized PRPE from Croatina grape (**a**) between 500 and 400 cm^−1^; (**b**) focus on the peak related to the inorganic phase (500–1500 cm^−1^); (**c**) focus on the phenolic phase (1450–1800 cm^−1^).

**Figure 4 jfb-12-00031-f004:**
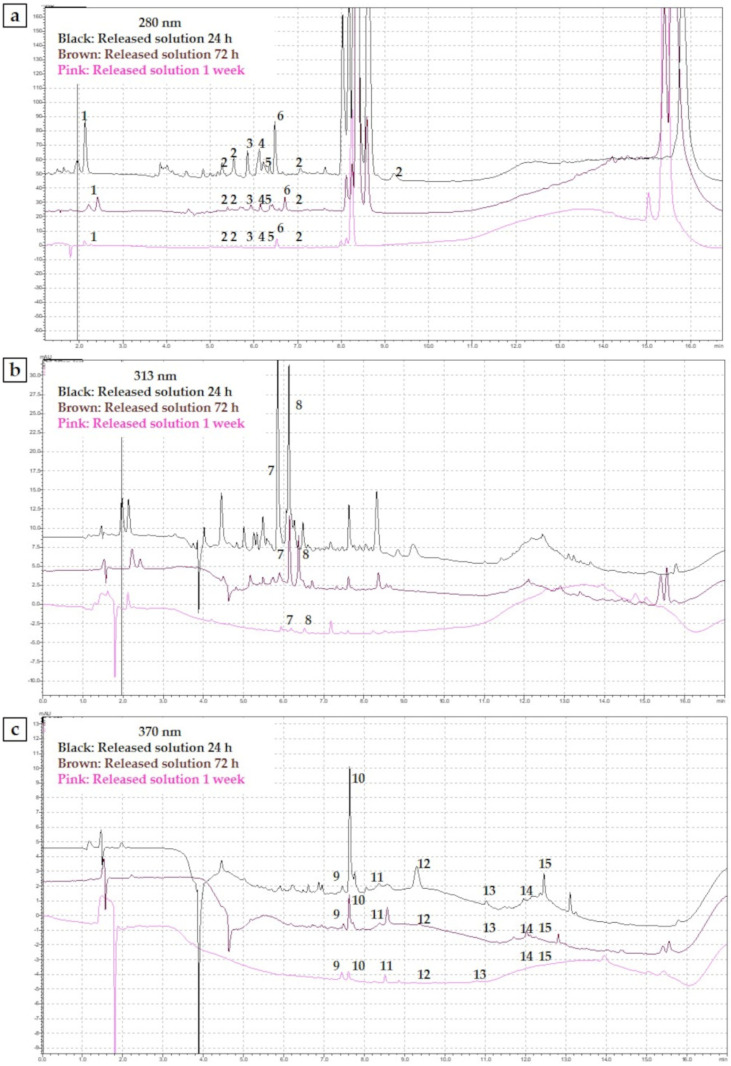
UHPLC analysis of the released solution after 24 h, 72 h, and 1 week. (**a**) Chromatograms at 280 nm; (**b**) Chromatograms at 313 nm; (**c**) Chromatograms at 370 nm. 1. Gallic acid, 2. Tannic acid, 3. Catechin, 4. Procyanidin B2, 5. Epicatechin, 6. Epigallocatechin gallate, 7. trans p-coumaroyl tartaric acid, 8. Caffeic acid, 9. Rutin, 10. Quercetin-3-glucuronide, 11. Quercitrin, 12. Myricetin, 13. Quercetin, 14. Kaempferol, 15. Isorhamnetin.

**Figure 5 jfb-12-00031-f005:**
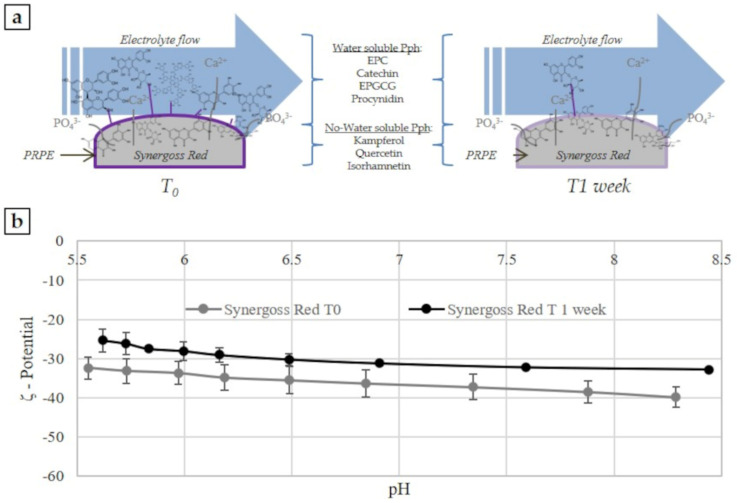
ζ-potential measurement of Synergoss Red before the release study and after 1 week in aqueous environment. After 1 week, a high percentage of flavonol molecules with reduced solubility in water can be observed on the surface of ceramic granules.

**Figure 6 jfb-12-00031-f006:**
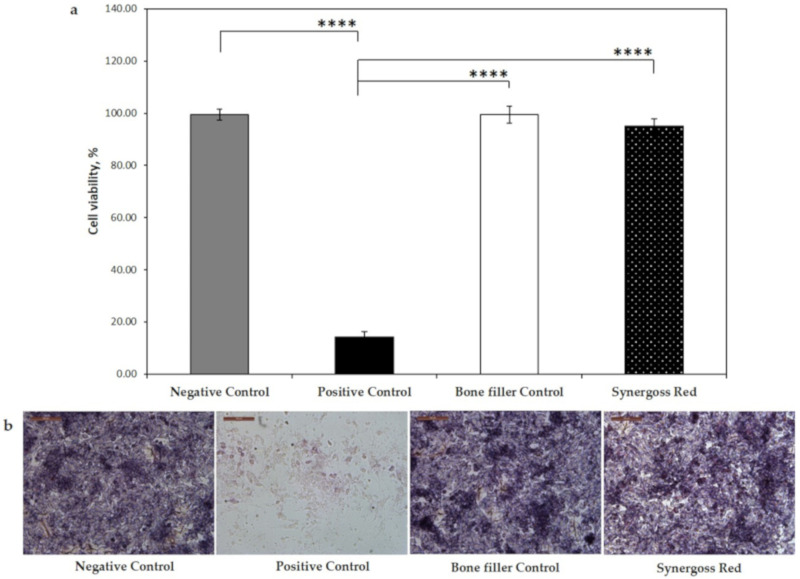
Cell viability assay of Saos-2 cells incubated in direct contact with Synergoss Red at the concentration of 0.2 g/well. (**a**) Cell viability of osteoblast-like cells in contact with Synergoss Red; (**b**) optical images of MTT stained osteoblast-like cells. The controls are represented by cells incubated without any biomaterial (negative control), with NPS (positive control), and with the bone filler without the pomace extract (bone filler control). Data are expressed as mean ± SD (n = 3). p values ≤ 0.05 were considered statistically significant (**** *p* ≤ 0.0001).

**Figure 7 jfb-12-00031-f007:**
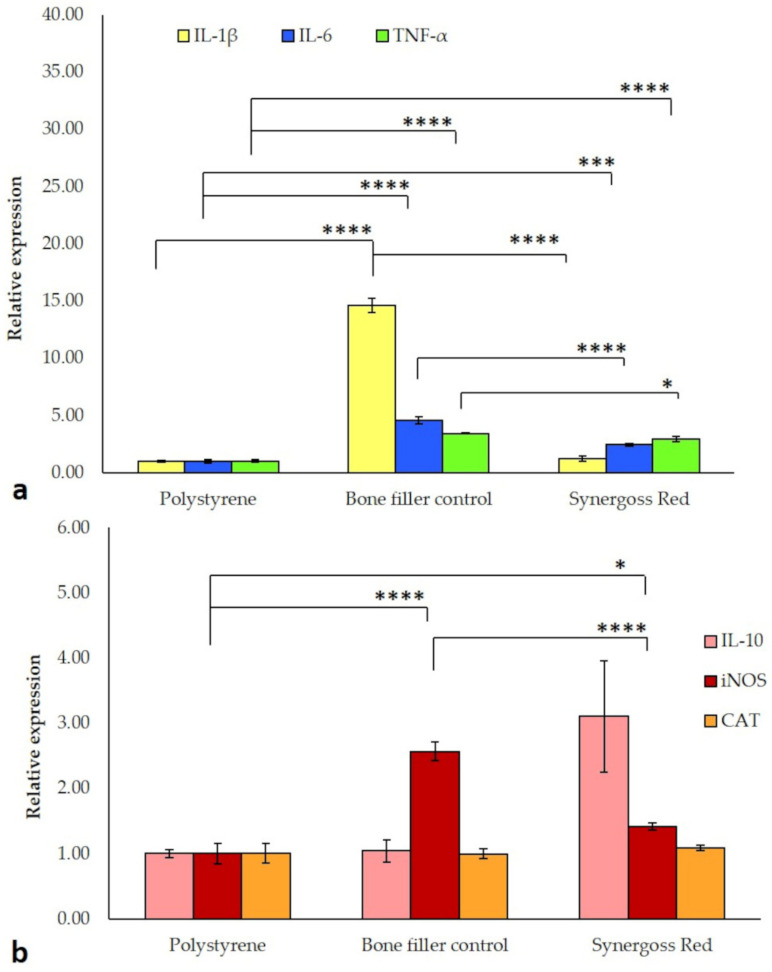
RT-qPCR analysis of genes involved in the inflammatory and oxidative stress response in J774a.1 macrophage-like cells incubated for 4 h in direct contact with 0.2 g/well bone filler control or Synergoss Red. (**a**) Anti-inflammatory action of Synergoss Red exerted through downregulation of pro-inflammatory genes. (**b**) The combined anti-inflammatory and anti-oxidant effects of Synergoss Red were able to reduce the expression of iNOS. Data are expressed as mean ± SD (n = 3). *p* values ≤0.05 were considered statistically significant (* *p* ≤ 0.05; *** *p* ≤ 0.001; **** *p* ≤ 0.0001).

**Figure 8 jfb-12-00031-f008:**
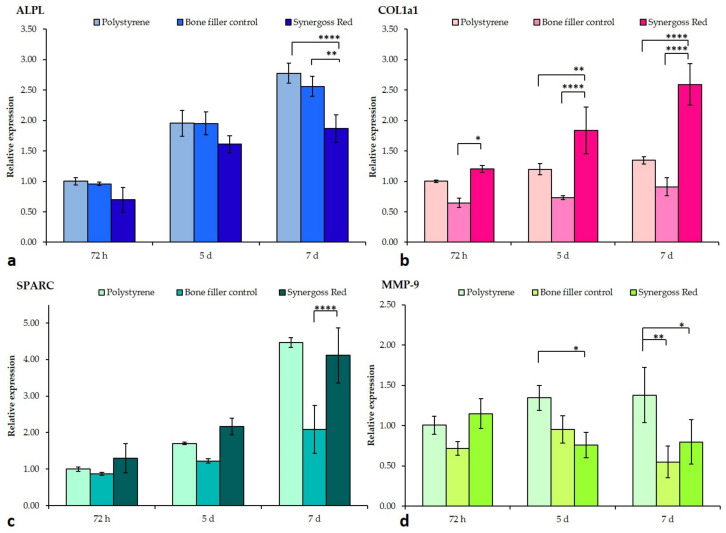
RT-qPCR analysis of genes involved in bone regeneration and tissue remodeling in SaOS2 osteoblast-like cells following incubation for 72 h, 5 d, and 7 d in direct contact with 0.2 g/well bone filler control or Synergoss Red. (**a**–**c**) Contribution of Synergoss Red in modulating the deposition of an early bone matrix; (**d**) Synergoss Red also modulated expression of genes encoding for proteinases such as MMP-9, which are responsible for ECM degradation. Data are expressed as mean ± SD (n = 3). *p* values ≤ 0.05 were considered statistically significant (* *p* ≤ 0.05; ** *p* ≤ 0.01; **** *p* ≤ 0.0001).

**Figure 9 jfb-12-00031-f009:**
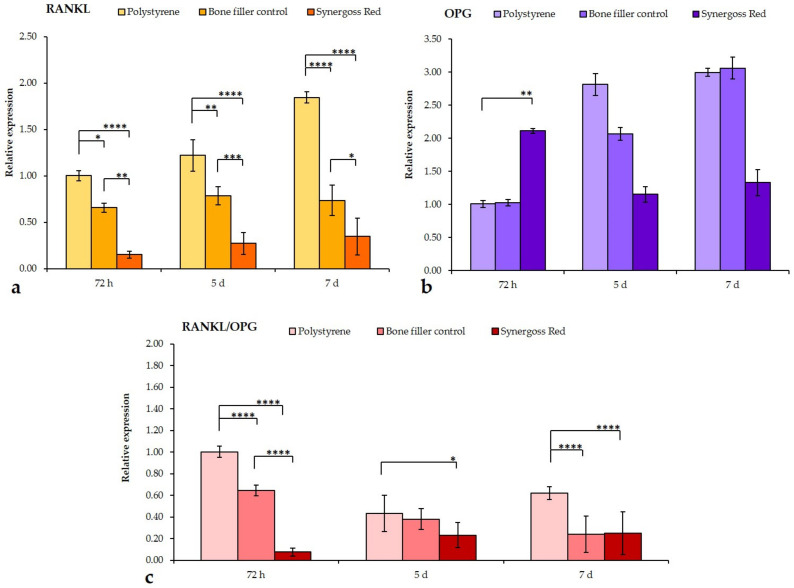
RT-qPCR analysis of genes involved in osteoclastogenesis and bone remodeling in SaOS2 osteoblast-like cells following incubation for 72 h, 5 d, and 7 d in direct contact with 0.2 g/well bone filler control or Synergoss Red. (**a**–**c**) Synergoss Red is able to downregulate the RANKL/OPG mRNA ratio thanks to its anti-inflammatory properties, which induced a considerable decrease of RANKL expression. Data are expressed as mean ± SD (n = 3). *p* values ≤ 0.05 were considered statistically significant (* *p* ≤ 0.05; ** *p* ≤ 0.01; *** *p* ≤ 0.001; **** *p* ≤ 0.0001).

**Table 1 jfb-12-00031-t001:** Antioxidant power and total phenolic content values from different batches of extract from the same grape pomace variety (Croatina) obtained from the same producer in the same year of production. No significant differences were found among batches.

Batch	Total Phenolic Content (GAE mg/mL)	Antioxidant Power (%)
TC CR1803	3.89 ± 0.17	61.2 ± 4.3
TC CR1901	3.939 ± 0.096	65.4 ± 5.7
TC CR2002	3.17 ± 0.15	72.8 ± 4.8

**Table 2 jfb-12-00031-t002:** Details of the HPLC gradient method used for the PRPE analysis.

Time (min)	MPA (%)	MPB (%)
0–3	100–70	0–30
3–8	70–40	30–60
8–11	40–100	60–100
11–13	0	100
13–14	0–70	100–30
14–16	70–100	30–0
16–17	100	0

**Table 3 jfb-12-00031-t003:** Surface composition (at. %) of the tested bone fillers analyzed by XPS.

Sample	O	C	Ca	P
Synergoss Red	36.2	53.8	3.8	6.2
BF	30.7	33.9	22.0	13.4

**Table 4 jfb-12-00031-t004:** Phenolic content and antioxidant power of the released solution in vitro at 24 h, 72 h, and 1 week.

Time	Total Phenolic Content (GAE mg/mL)	Antioxidant Power (%)
24 h	0.550 ± 0.012	8.74 ± 0.16
72 h	0.135 ± 0.026	1.26 ± 0.48
1 week	0.0376 ± 0.0072	0.47 ± 0.37

**Table 5 jfb-12-00031-t005:** Quantification of polyphenols through UHPLC analysis of the PRPE and released solution (after 1 week). The percentage of the released polyphenols from the polyphenols identified and quantified in the initial PRPE through UHPLC was calculated.

Polyphenols (Pph)	Pph in PRPE (µg/mL)	Pph in Total Released Solution (µg/mL)	% of Pph Released
Gallic Acid	33.0 ± 4.0	22.0 ± 3.9	66.6
Quercetin 3-glucuronide	46.2 ± 1.3	2.89 ± 0.82	6.3
Quercetin	62 ± 11	10.57 ± 0.53	17.1
Tannic Acid	1086 ± 160	136 ± 22	12.6
Catechin	71.0 ± 2.0	26.0 ± 4.0	36.4
Malvidin 3-glucoside	2.04 ± 0.75	0.173 ± 0.058	8.5
Epicatechin	55.7 ± 4.3	20.9 ± 2.4	37.6
Procyanidin B2	98.1 ± 5.8	37.1 ± 3.7	37.8
Caffeic acid	15.99 ± 0.25	2.21 ± 0.55	13.8
Myricetin	13.9 ± 2.9	3.0 ± 0.5	21.6
Quercitrin	103.4 ± 5.1	2.28 ± 0.48	2.2
Kaempferol	38.4 ± 6.7	2.14 ± 0.22	5.6
Isorhamnetin	74 ± 21	0.660 ± 0.099	0.9
Rutin	40.8 ± 2.9	0.646 ± 0.013	1.6
Epigallocatechin gallate	44.4 ± 1.2	21.0 ± 3.2	47.2
Trans p-coumaroyl tartaric acid	23.48 ± 0.69	5.7 ± 1.40	24.1

## Data Availability

Not applicable.

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
