# Peer review of "Functionalization with a Polyphenol-Rich Pomace Extract Empowers a Ceramic Bone Filler with In Vitro Antioxidant, Anti-Inflammatory, and Pro-Osteogenic Properties"

_jfb, 2021, doi:10.3390/jfb12020031_

Round 1
Reviewer 1 Report
The present study reports the biological effects on bone cells in response to the functionalization of ceramic with a polyphenol-rich pomace extract. The article is well written and the objectives are clearly identified.
Several points need to be addressed :
- reproducible content of extracts should be further documented as it seems dependant on the weather, the quality of the soil in terms of pH, salinity...
- Table 1 is difficult to analyse as it lacks the meaning of the abreviations used (FC ...).
- the authors should also include analysis of residual solvents in the extract.
- methods of incorporating extract into ceramic must be more detailed.
- the authors should comment the type of interactions between the extract and the ceramic (i.e physical adsorption instead of chemical bonding...)
- regarding the release profile, more time points must be studied to reach more than 75% of release for exemple (ideally to get close to 100%)
- In general, physical adsorption of molecules provides a fast release mostly in the first day. How did the authors explain the low release rate of DPPH in table 4?
- the authors should clarify the difference between data obtained in table 4 vs table 5 as % of molecules released is strongly different.
- Did the authors evaluate the in vitro effect of pomace extract alone in solution ? If yes, did they adjust the rate incorporation of pomace extract from the results obtained ? It should be further discussed.
Reviewer 2 Report
Thank you for giving this opportunity to review novel manuscript. This is a manuscript on the in vitro performance of HA/beta-TCP scaffold soaked in polyphenol-rich pomace extracts. This manuscript was little bit too long, but handle very interesting topic. It was nicely written and included a lot of material characterization and in vitro evaluation. The Synergross Red was not cytotoxic, but increase osteogenic markers. This manuscript was so impressive there are only few to point out.
I would like to see the results of in vivo study using this material in near future, which would be included in the next manuscript. I am looking forward to the next paper regarding this interesting material.
Minor issue
Line 37
What was the definition of success here? No infection or loosening?
Figure1. what was the difference in SEM images between the control and Synergross Red? Did authors find any topographical change in SEM images after soaking the material in Polyphenols rich pomace extract?
Line 758
The presence of polyphenols on the surface of granulates was confirmed by zeta potential analysis.
Is this most convincing way to prove the presence of polyphenols? If no, authors should just state “suggested” or “implied” instead of “confirmed”.
